# Combined reference-free and multi-reference based GWAS uncover cryptic variation underlying rapid adaptation in a fungal plant pathogen

Anik Dutta[1¤], Bruce A. McDonald[1], Daniel Croll [2]*

1 Plant Pathology, Institute of Integrative Biology, ETH Zurich, Zurich, Switzerland, 2 Laboratory of Evolutionary Genetics, Institute of Biology, University of Neuchâtel, Neuchâtel, Switzerland

¤ Current address: Institute of Phytopathology, Christian-Albrecht University of Kiel, Kiel, Germany
* daniel.croll@unine.ch

**Data Availability Statement:** All genome sequences are available from the NCBI Sequence Read Archive (BioProject accessions

## Abstract

Microbial pathogens often harbor substantial functional diversity driven by structural genetic variation. Rapid adaptation from such standing variation threatens global food security and human health. Genome-wide association studies (GWAS) provide a powerful approach to identify genetic variants underlying recent pathogen adaptation. However, the reliance on single reference genomes and single nucleotide polymorphisms (SNPs) obscures the true extent of adaptive genetic variation. Here, we show quantitatively how a combination of multiple reference genomes and reference-free approaches captures substantially more relevant genetic variation compared to single reference mapping. We performed reference-genome based association mapping across 19 reference-quality genomes covering the diversity of the species. We contrasted the results with a reference-free (i.e., k-mer) approach using raw whole-genome sequencing data in a panel of 145 strains collected across the global distribution range of the fungal wheat pathogen *Zymoseptoria tritici*. We mapped the genetic architecture of 49 life history traits including virulence, reproduction and growth in multiple stressful environments. The inclusion of additional reference genome SNP datasets provides a nearly linear increase in additional loci mapped through GWAS. Variants detected through the k-mer approach explained a higher proportion of phenotypic variation than a reference genome-based approach and revealed functionally confirmed loci that classic GWAS approaches failed to map. The power of GWAS in microbial pathogens can be significantly enhanced by comprehensively capturing structural genetic variation. Our approach is generalizable to a large number of species and will uncover novel mechanisms driving rapid adaptation of pathogens.

## Author summary

Mapping trait variation to polymorphism within species has become a cornerstone of modern biology. Applications in microbial pathogens (both bacteria and fungi) have

PRJNA327615, PRJNA596434, and
PRJNA178194).

**Funding:** BAM was supported by the Swiss Federal
Office for Agriculture (BLW) in the framework of
the NAP-PGREL Project Nr. 627000640. The
funder had no role in study design, data collection
and analysis, decision to publish, or preparation of
the manuscript.

**Competing interests:** The authors have declared
that no competing interests exist.

produced major insights into their ecology, emergence of resistance, gains in virulence
and climatic adaptation. However, microbial populations collected from the environment
often express major adaptive traits governed by complex genetic variation. Standard
genome-wide association studies (GWAS) based on single nucleotide polymorphisms
have typically failed to reveal the full extent of loci contributing to such traits. We provide
the first quantitative assessment of GWAS in a fungal pathogen comprehensively account-
ing for complex sequence variation. We used an environmental collection of the global
fungal pathogen of wheat, *Zymoseptoria tritici*, as a case study. We analyzed a panel of 145
strains collected across the global distribution range and gathered trait variation data on
49 distinct traits. We integrated complex sequence variation among strains systematically
into the association mapping and found that multiple approaches are needed to cover sat-
isfactorily causal loci for trait variation. Our approach is generalizable to many microbial
species and will uncover novel mechanisms driving rapid host adaptation in microbial
populations.

## Introduction

Rapid genetic change in plant pathogens has led to significant damage to agricultural produc-
tion over recent decades [1–3]. The rapid evolution in pathogen populations of virulence and
resistance to anti-microbial drugs are key concerns. There is an urgent need to identify the
precise genetic determinants in pathogens that underlie differences in virulence and evasion of
control mechanisms. Vast genomic datasets can now be exploited to retrace evolutionary path-
ways of pathogen adaptation. Association mapping can be used to establish relationships
between genetic and phenotypic variation using field collections of pathogens [4–6]. The
genetic variation relevant for trait evolution is often more complex than the commonly used
single nucleotide polymorphisms (SNPs). Structural variants (SVs) such as insertions-dele-
tions (indels), copy number variants, chromosomal rearrangements, inversions and duplica-
tions can also be major facilitators of microbial adaptation [7–11]. For plant studies, powerful
approaches were recently proposed to associate SVs to causal genes controlling trait variation
[12,13]. However, our understanding of SVs governing trait variation in microorganisms is
limited by approaches focused on SNPs [14–17]. Microbial genomes are highly plastic in terms
of gene content and associated SVs. GWAS based on a single reference-genome can only cap-
ture the gene content described in that single genome [18]. Using a compilation of reference
genomes to construct a pangenome resource that integrates a more comprehensive set of the
genes present in a pathogen species shows substantial promise [19]. The ability to integrate
various types of SVs while performing association mapping will also substantially expand our
understanding of microbial adaptation.

Pathogen adaptation is frequently governed by genetic determinants termed accessory
genes that are not shared among all individuals of a species. Accessory genes were found to
affect defense responses, virulence, drug resistance and environmental adaptation [6,20–22].
The detection of such adaptive accessory genes can be accelerated by expanding GWAS to
include multiple reference genomes covering distinct segments of the gene space of a species.
Additionally, single reference genome based GWAS can be confounded by gene presence/
absence variation as such variation is challenging to account for [23]. These shortcomings of a
GWAS based on a single reference genome can be overcome by repeating the mapping across
multiple reference genomes representing the pangenome of a species [24–26]. Recent advances
in genomics are rapidly expanding the number of microbial pathogens with such pangenome

resources [27–31]. These resources can facilitate the identification of pathogen virulence factors as well as previously unknown anti-microbial resistance factors emerging after the application of newly designed chemical control agents [11,32]. In particular, SVs in highly repetitive regions are unlikely to be captured. This can be overcome by adopting an alignment-free approach where short reads are screened for subsequences of specific length, *i.e.* k-mers [33,34]. A major advantage of k-mer based analyses is the ability to capture genetic variation without depending on a reference genome, avoid SNP calling ascertainment biases or allow identifying sequence segments absent from a reference genome [18,35,36]. Capturing complex SVs is particularly relevant because significant genetic variation, sometimes referred to as the "missing heritability" problem, can go undetected using traditional reference-based GWAS [37,38]. Though their potential advantages are clear and have been applied in several bacterial and plant studies [39–43], reference-free methods to capture adaptive genetic variation remain largely unexplored in plant pathogenic microorganisms.

The fungal pathogen *Zymoseptoria tritici* causes septoria tritici blotch (STB), a disease that has a significant impact on global wheat production [44,45]. *Z. tritici* has a highly plastic genome with 13 core chromosomes and 8 accessory chromosomes that exhibit presence-absence variation among isolates [46]. Large effective population sizes, high gene flow and high recombination rates facilitate rapid evolution of resistance toward fungicides and virulence on resistant hosts [47–50]. The pathogen population harbors substantial variation for many life history traits including growth rates, stress tolerance, melanization and reproduction on the wheat host [51]. Structural rearrangements and deletion events were found to be associated with host adaptation [52,53]. GWAS based on single reference genomes was successful in discerning the genetic underpinnings of pathogen virulence and fungicide resistance [14,47]. The recent pangenome constructed for *Z. tritici* based on 19 different isolates from six continents showed that the pathogen harbors a substantially larger gene repertoire than the canonical reference genome [30]. Accessory genes within the species encode diverse but largely unknown functions and were likely missed in previous analyses that relied on a single reference genome. Thus, expanding GWAS beyond one reference genome will likely capture a larger fraction of genes underlying recent adaptation.

Here, we assess the performance of both reference-free and multi-reference GWAS by conducting a comprehensive mapping analysis based on a global set of *Z. tritici* populations. We screened for sources of genetic variation affecting 49 biotic and abiotic traits. Both GWAS conducted on SNP datasets mapped to 19 different reference genomes and k-mer based GWAS revealed a large number of previously missed loci contributing to trait variation. Our study provides quantitative insights into how improved GWAS approaches can identify genetic variants underpinning adaptation in rapidly evolving microbial pathogens.

## Results

### A generalizable framework for conducting microbial GWAS

We performed comprehensive association mapping analyses to detect genetic variants of varying complexity underlying pathogen adaptation to different hosts and environments (**Fig 1**). We analyzed genetically diverse pathogen populations spanning the global distribution of wheat and recapitulating host diversity and climatic gradients. Isolates were phenotyped under greenhouse and laboratory conditions to assess both pathogenicity-related traits (e.g., degree of host damage, amount of spore production) and responses to abiotic stresses (e.g., fungicide, low temperature) [51]. Genetic variation in the mapping panel was assessed in two complementary ways. (1) Whole-genome sequence datasets were used to generate SNP calls on multiple reference genomes. A total of 19 telomere-to-telomere reference genomes have been

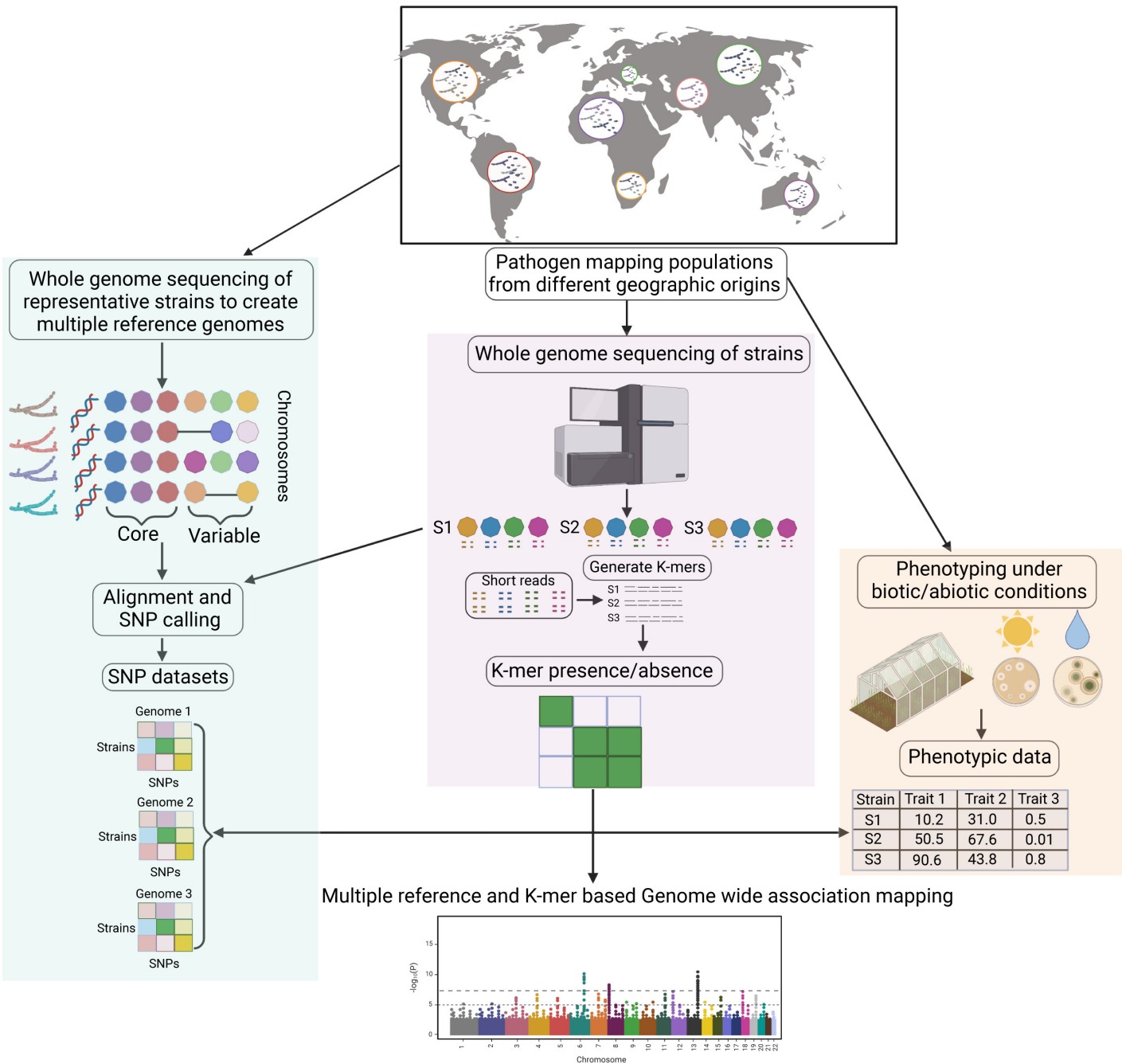

**Fig 1. A comprehensive workflow for conducting microbial genome-wide association studies (GWAS) using multiple reference genomes and k-mer data from mapping populations.** Genetically diverse pathogen populations from different geographic locations are sampled to construct an association panel followed by greenhouse and laboratory phenotyping to assess heritable trait variation (right panel; [51]). Chromosome-level genome assemblies of representative isolates is performed to generate reference genomes and establish a species pangenome (left panel; [30]). Whole genome sequencing of the association panel enables single nucleotide polymorphism (SNP) calling against multiple reference genomes and creation of k-mer presence/absence tables (middle panel). GWAS can be performed simultaneously to take advantage of SNP datasets or k-mer presence/absence tables. The icons were adopted from biorender.com. The map was created using *ggmap* [98] in R with map data from OpenStreetMap: https://www.openstreetmap.org/copyright.

assembled to capture the global diversity in structural variation [30]. (2) Short reads were also used to generate 25-bp k-mer profiles for each isolate. These presence/absence k-mer tables applied to mapping populations are highly effective in capturing structural variation independent of a reference genome [35].

## Multiple reference genome based GWAS

We performed association mapping for a total of 49 traits including measures of virulence and reproduction on twelve wheat cultivars and growth and melanization under various stress conditions such as different temperature regimes and fungicide exposure (S1A Table). The mapping was performed independently for SNP panels generated from each of the 19 reference genomes based on mixed linear models. We estimated the genomic inflation factor for each reference genome (GIF; λ), which ranged on average from 0.91 to 1.09 without principal components as a random effect controlling for population substructure, and on average from 0.70 to 1.36 when including the first three principal components as covariates (S1 Fig). We noticed $p$-value inflation when using principal components as covariates (S2 Fig). Consequently, all reported significant SNP associations were obtained by conducting GWAS without principal components as covariates. The multiple reference-based GWAS detected a range of significant marker-trait associations above the Bonferroni threshold ($\alpha = 0.05$) for a total of 20 traits related to virulence, reproduction, growth rate, fungicide resistance and melanization out of 49 total traits tested (Fig 2A; S1A and S1B Table). Therefore, for the following we focus only on GWAS outcomes for these 20 traits. We observed variability in the number of significant SNPs associated with the same trait according to the chosen reference genome SNP panel (Fig 2A, S1B Table). We computed the coefficient of variation (CV) to standardize the variability across the 19 reference genomes for each trait. The traits with a high numbers of significant SNPs such as fungicide resistance, reproduction on wheat cultivar Titlis, virulence on landraces 1204 and 4391 showed comparatively lower CV than traits with a small number of significant SNPs. The number of significant SNPs ranged from 1–55 for pathogen virulence and reproduction on different wheat hosts depending on the reference genome. The highest number of significant SNPs were identified for virulence on landrace 1204 with the alternative reference genome KE94 (Fig 2B). This trait also showed the highest variance in the number of significant associations among the 19 reference genomes (S1B Table). The number of significant SNPs for environmental stresses ranged from 1–180 with the azole resistance trait showing the largest and most variable number of SNPs among the 19 reference genomes. The most significant SNPs for each trait explained 3–15% of the phenotypic trait variation (S1C Fig). This suggests that numerous genes affect most trait variation in most environments, consistent with polygenic architectures for most of these traits.

A substantial fraction of all significant associations could not be mapped with the canonical reference genome IPO323 (Figs 2B and S3). For example, our GWAS analysis on the KE94 reference genome identified several significant SNPs associated with a gene encoding a fungal-specific transcription factor. These SNPs associations were notably absent in the canonical reference genome IPO323. Further analysis revealed that several SNPs localized in the promoter region of the gene in KE94 were not discovered in IPO323. Depending on the local pattern of linkage disequilibrium (LD), it is plausible that the significant SNPs in KE94 are in LD with the SNPs in the promoter region (S3 Fig). Also, significant associations for several traits mapped in to the canonical reference genome were not found using alternative reference genomes (Fig 2B). This shows that multiple reference genome SNP panels can overcome limitations due to presence-absence variation and challenges in SNP calling. To analyze putative gene functions contributing to phenotypic trait variation, we extracted all the genes in close

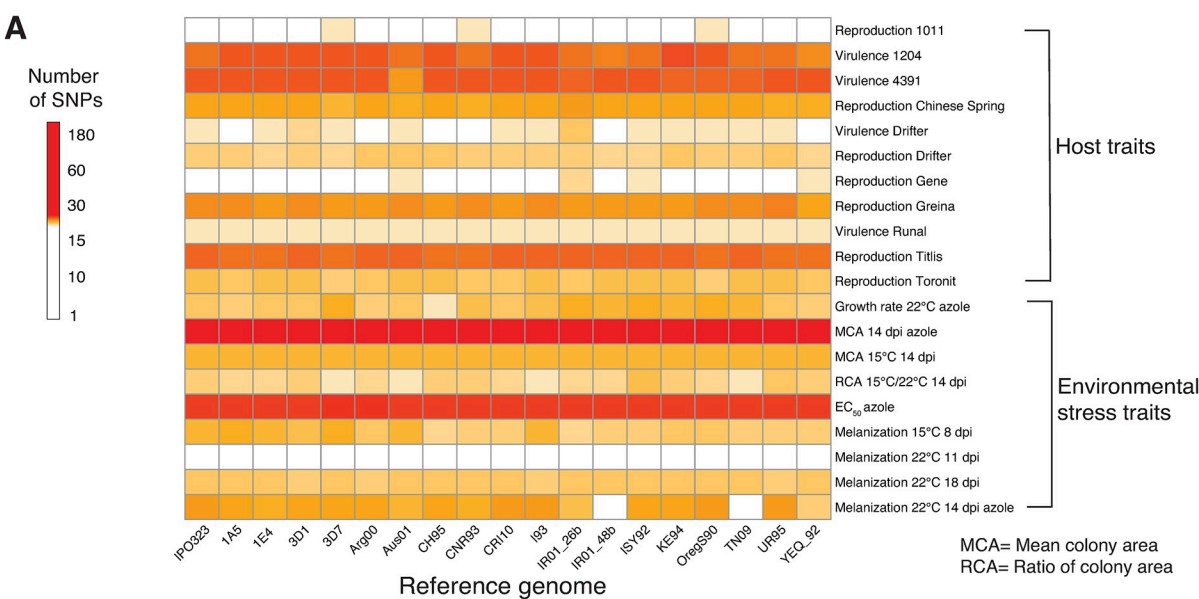

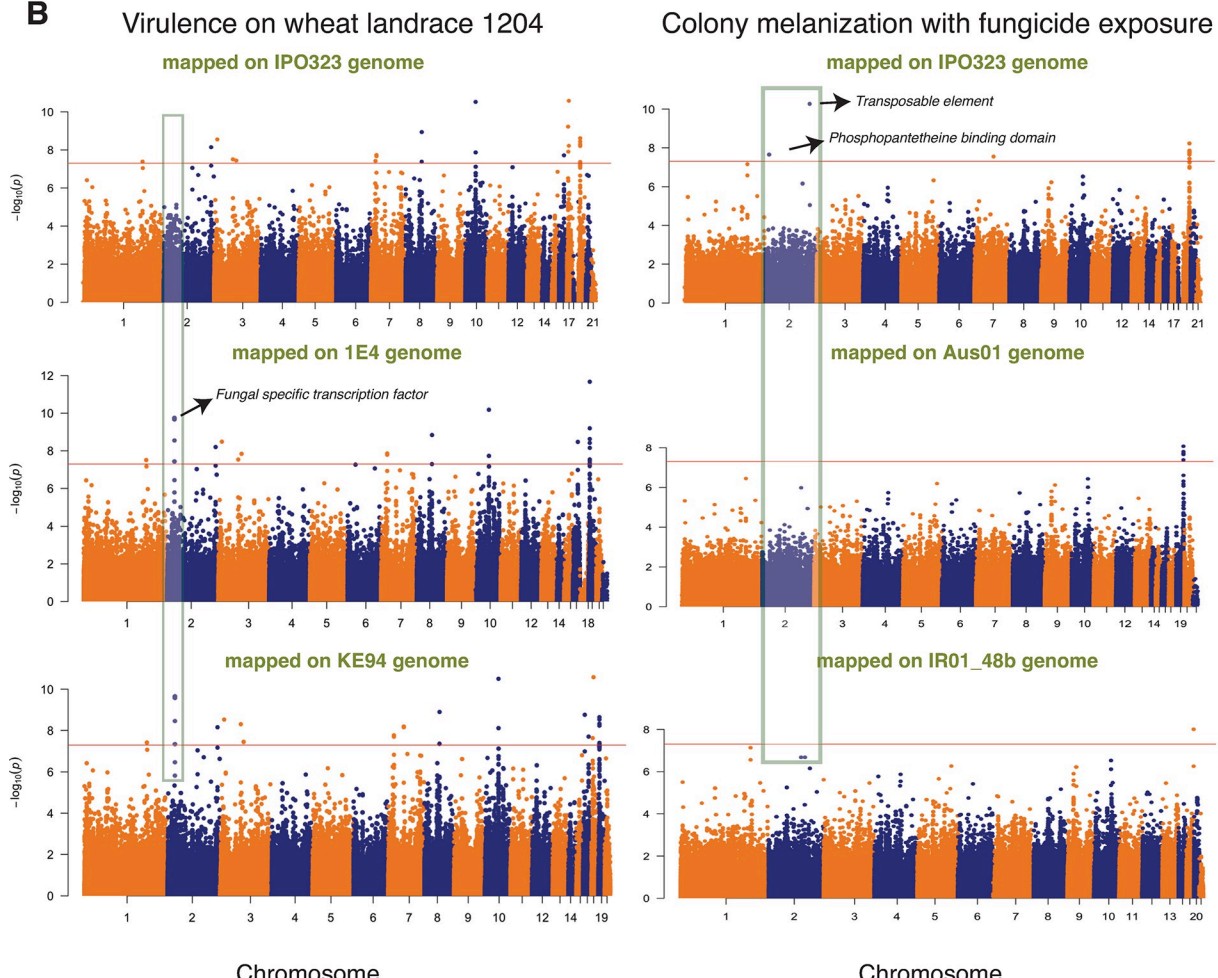

**Fig 2. Genome wide association mapping based on 19 reference genomes for 49 pathogen traits measured under different host and abiotic conditions in *Zymoseptoria tritici*.** (A) Heatmap showing the impact of reference genome selection on the number of significantly

associated SNPs in GWAS for each trait. Pathogen virulence (percentage of the leaf surface covered by necrotic lesions) and reproduction (pycnidia density within lesions) were measured on 12 genetically diverse wheat lines. (B) Manhattan plots showing SNP *p*-values for two traits (pathogen virulence in the left panel and melanization in presence of fungicide in the right panel) on multiple reference genomes. The shaded gray boxes highlight differences in significant associations found when using different reference genomes. The red line indicates the Bonferroni threshold at a 5% significance level.

physical proximity to each SNP (< 1 kb). *Z. tritici* populations show rapid decay in linkage disequilibrium within this distance and the average distance between genes is ~1 kb [46,52]. We identified a variable number of associated genes depending on the reference genome SNP panel. The number of associated genes ranged from 54 when mapping was performed on the reference genome Aus01 to 79 on IPO323 for pathogen virulence and reproduction on different wheat hosts. Each gene was counted only once even if the gene was mapped as significantly associated for multiple traits. The number of genes ranged from 88 (reference genome TN09) to 102 (reference genome CRI10) for environmental stress traits (*i.e.* fungicide resistance, growth rate and melanization; **S1D Table**). Based on the annotation of the canonical reference genome IPO323, the identified genes encoded a broad range of functions including major facilitator superfamily (MFS) transporters, fungal-specific transcription factors, zinc finger and protein kinase domains (**S1D Table**). Such gene functions may have specific metabolic and regulatory functions underlying pathogen adaptation [17,54]. Importantly, we detected significant SNPs near three genes encoding predicted virulence factors (*i.e.* effectors) on chromosomes 2, 5, and 7 associated with reproduction on the wheat cultivars Greina, Titlis and Chinese Spring, respectively (**S1D Table**). We also detected numerous significant SNPs for azole resistance tagging the *CYP51* gene that is known to underlie resistance to azole fungicides [55].

A challenge associated with performing multiple reference genome GWAS is to identify redundant associations across SNP panels. To estimate the extent of novel gene functions discovered through the expansion of the reference genome SNP panels, we performed a saturation analysis based on orthology information. For each gene with a significant association, we assessed whether any ortholog identified in a different reference genome was already tagged (*i.e.* is a member of the same orthogroup). We randomly selected subsets of the reference genome SNP panels and counted the number of unique orthogroups with significant associations for groups of traits. We observed a sub-linear increase in the number of unique orthogroups with significant associations with an increasing number of reference genome panels (**Fig 3**). The most substantial increase was observed by including a second reference genome panel. Beyond two reference genome panels, the benefits for each additional reference genome SNP panel decreased slightly. This shows that a substantial fraction of the genetic factors contributing to adaptation to host, and environmental stress factors cannot be identified from a single reference genome SNP panel. Fungicide resistance related traits show the highest number and fastest gain in significantly associated orthogroups with additional reference genome SNP panels. Pathogen virulence and reproduction showed intermediate increases in significantly associated orthogroups and melanization showed the slowest increase in significantly associated orthogroups. Overall, including multiple reference genome SNP panels substantially expands the spectrum of identifiable genetic factors (**S6 Fig**).

## k-mer approach to uncover additional sources of genetic variation

To further expand our survey of structural variation potentially associated with trait variation, we performed reference-free GWAS on the same trait dataset using 25-bp k-mers generated from whole genome sequencing data. Considering the small genome size of *Z. tritici* (~40

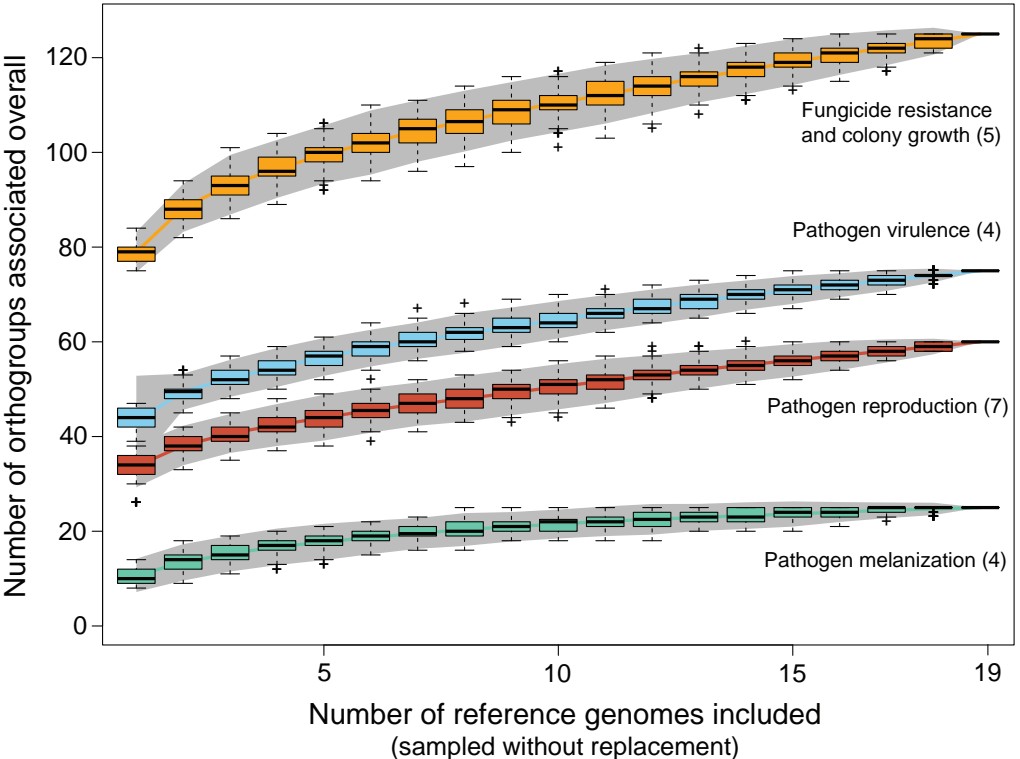

**Fig 3. Accumulation curves for the total number of distinct genes (identified by orthogroups within the species) associated with GWAS for different traits as a function of the number of reference genomes analyzed.** Mapping outcomes are shown for different groups of traits. The number of distinct genes across these groups of traits were summed. The numbers in parentheses indicate the number of traits included in each category. Pathogen virulence (percentage of the leaf surface covered by necrotic lesions) and reproduction (pycnidia density within lesions) were measured on 12 genetically diverse wheat lines.

Mb), we chose to cap k-mer length at 25-bp as this will keep numbers of unique k-mers reasonably low, which reduces the computational burden and sensitivity to minor mutations that could prevent k-mers being mapped to specific positions in a reference genome. We identified a total of ~55 million k-mers of which 7,111,640 were detected in at least five isolates. We estimated k-mer based heritability to contrast with SNP-based heritability from [51]. We created separate genetic relatedness matrices of all genotypes for each reference genome and all the k-mers tables. Therefore, the heritability estimates directly reflect relatedness among individuals based on either SNPs or k-mers. For pathogen virulence, k-mers explained a higher proportion of phenotypic variance compared to the SNP-based estimates (**Fig 4A**). A similar trend of increased heritability accounted by k-mers was observed for all other traits as well (**Figs S4A, 4B and 4C**). The heritability for virulence ranged from 0 to 0.84 (standard error, SE = 0.08) with an average of 0.6 (SE = 0.16) compared to 0.35 (SE = 0.14) based on SNPs. Heritability for reproduction traits ranged from 0.73 (SE = 0.13) to 0.96 (SE = 0.01) with an average of 0.86 (SE = 0.06) compared to SNP-based heritability with an average of 0.65 (SE = 0.1). The average heritability for environmental stress factors (i.e., fungicide resistance, growth rate and melanization at different temperatures) was 0.7 (SE = 0.18) compared to 0.51 based on SNPs (SE = 0.18). Consistent with the high heritability estimates, the k-mer GWAS yielded numerous k-mers above the permutation-based significance threshold ($\alpha = 0.05$) for 33 out of 49 phenotypic traits. The number of significant k-mers ranged from 3–2066 for pathogen

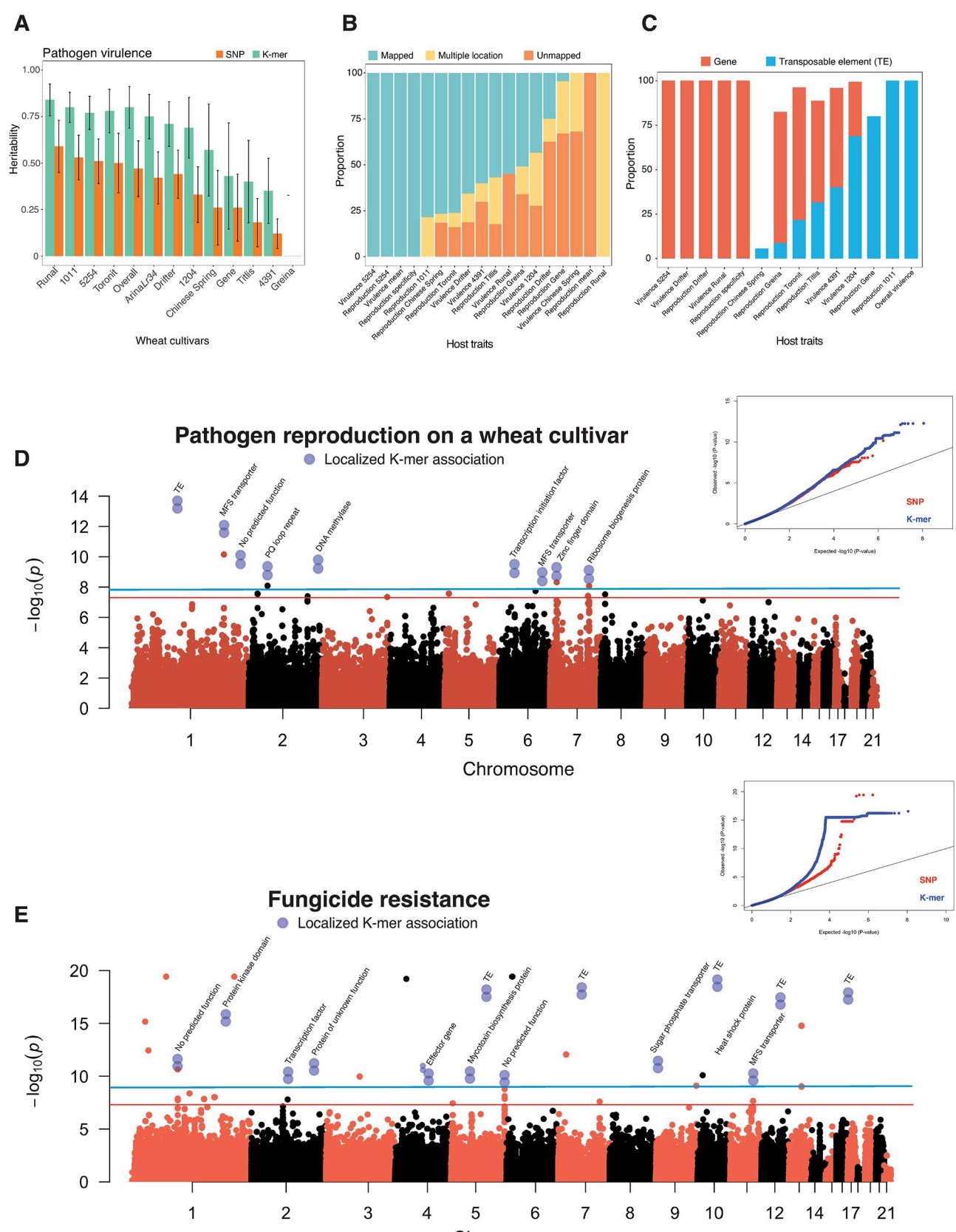

**Fig 4. k-mer GWAS on 49 life-history traits based on a k-mer presence/absence table for all 145 *Zymoseptoria tritici* isolates.** (A) Comparison of heritability estimates for pathogen virulence (percentage of the leaf surface covered by necrotic lesions) based on SNPs (for the reference genome IPO323) and k-mers. Both SNP-based and k-mer-based heritability were estimated by following a genome-based restricted maximum likelihood (GREML) approach. Standard errors are indicated by error bars (B) Alignment of significantly associated k-mers against the reference genome (IPO323) show the proportion of k-mers having a unique mapping position, multiple locations, or no unambiguous mapping position in host-related traits *i.e.* pathogen virulence and reproduction (pycnidia density within lesions). (C) Proportion of significant k-mers with a unique mapping position in the reference genome either tagging a gene or a transposable element for host-related traits. (D, E) Manhattan plots showing significant k-mer associations with pathogen reproduction and fungicide resistance together with quantile-quantile plots for *p*-value comparisons. Manhattan plots were created from SNP-based GWAS and blue dots represents the significant k-mer associations with the k-mers being uniquely mapped to a location in the reference genome. The two blue dots represent individual k-mers with significant associations. The blue dots in pairs indicate the presence of multiple k-mers association. The red and blue lines indicate the Bonferroni and permutation-based significance threshold at 5% level for SNPs and k-mers, respectively. Pathogen virulence and reproduction were measured on 12 genetically diverse wheat lines. Overall virulence and reproduction represent the average value of the respective trait measured on 12 genetically diverse wheat lines. Reproduction specificity was estimated based on the adjusted coefficient of variation of mean reproduction across 12 genetically diverse wheat lines. Higher specificity suggests affinity to certain hosts for maximizing reproductive fitness.

virulence, from 3–640 for pathogen reproduction, from 3–166 for pathogen melanization, and from 9–3606 for fungicide resistance and growth-related traits. An important strength of the k-mer-based GWAS is the ability to detect variation at loci not captured by reference genomes. Quality control steps during raw sequencing read cleaning is important though to minimize erroneous k-mers being included.

To identify gene functions mapped through k-mer GWAS, we searched k-mer sequences in the canonical reference genome IPO323 (**Figs 4B and S4D**). We observed heterogeneous patterns of significant k-mer mapping, including uniquely mapped k-mers to the canonical reference genome, as well as k-mers that mapped to multiple locations and k-mers that could not be mapped to the reference genome (**S4D Fig**). Since many significant k-mers could not be mapped to the IPO323 genome, we explored this issue by mapping significant k-mers to a second reference genome (*i.e.* 1A5). The proportion of k-mers with unique positions, multiple locations, and those that could not be mapped to the 1A5 reference genome followed a similar pattern than the mapping patterns on the canonical reference genome (**S5 Fig**). We found a substantial fraction of significant k-mers tagging either a transposable element (TE) or a gene in the *Z. tritici* genome (**Figs 4C and S4E**). For host-related traits (**Fig 4B**), an average of 63.6% of all significant k-mers tagged a gene compared to 32.1% tagging a TE. In contrast, the proportions of significant k-mers tagging a TE or a gene were roughly inverted (59.2% *vs.* 34.6%) for environmental stress traits (**S4E Fig**). Furthermore, for the majority of the traits, the k-mer with the highest *p*-value tagged a TE (**Fig 4D and 4E**). The high proportion of k-mers mapping to a TE suggests that active transposition has contributed significantly to phenotypic variation in *Z. tritici*. Additionally, the k-mer GWAS discovered a large number of not previously identified genes associated with both host-related and environmental stress traits (**Figs 4D and 4E and S6**). The k-mer tagged genes encoded a broad range of functions including a transcription factor, MFS transporters, and peptidases as well as effector candidates (**Fig 4D, 4E and S1E Table**). We incorporated the GRM as a random factor into the GWAS models to address the issue of cryptic genetic relatedness. Note that the observed inflation in *p-values* is most likely attributed to substantial population differentiation in phenotypic values such as the population from Switzerland exhibiting significantly higher resistance to fungicides compared to the other four populations.

We analyzed in detail how the k-mer approach expanded the discovery of loci compared to SNP-based GWAS. We focused on the key azole resistance gene *CYP51* (**Fig 5A**). We found 294 k-mers above the 5% significance threshold on chromosome 7 associated with *CYP51* (*Zt09_07_00450*) for the resistance trait EAM_14_dpi_azole. All the k-mers could be located to a unique position on the chromosome. The k-mer *p*-values tagging this gene were lower than the SNP *p*-values (**Fig 5B**). Nearly all (293/294) k-mers were located in the downstream

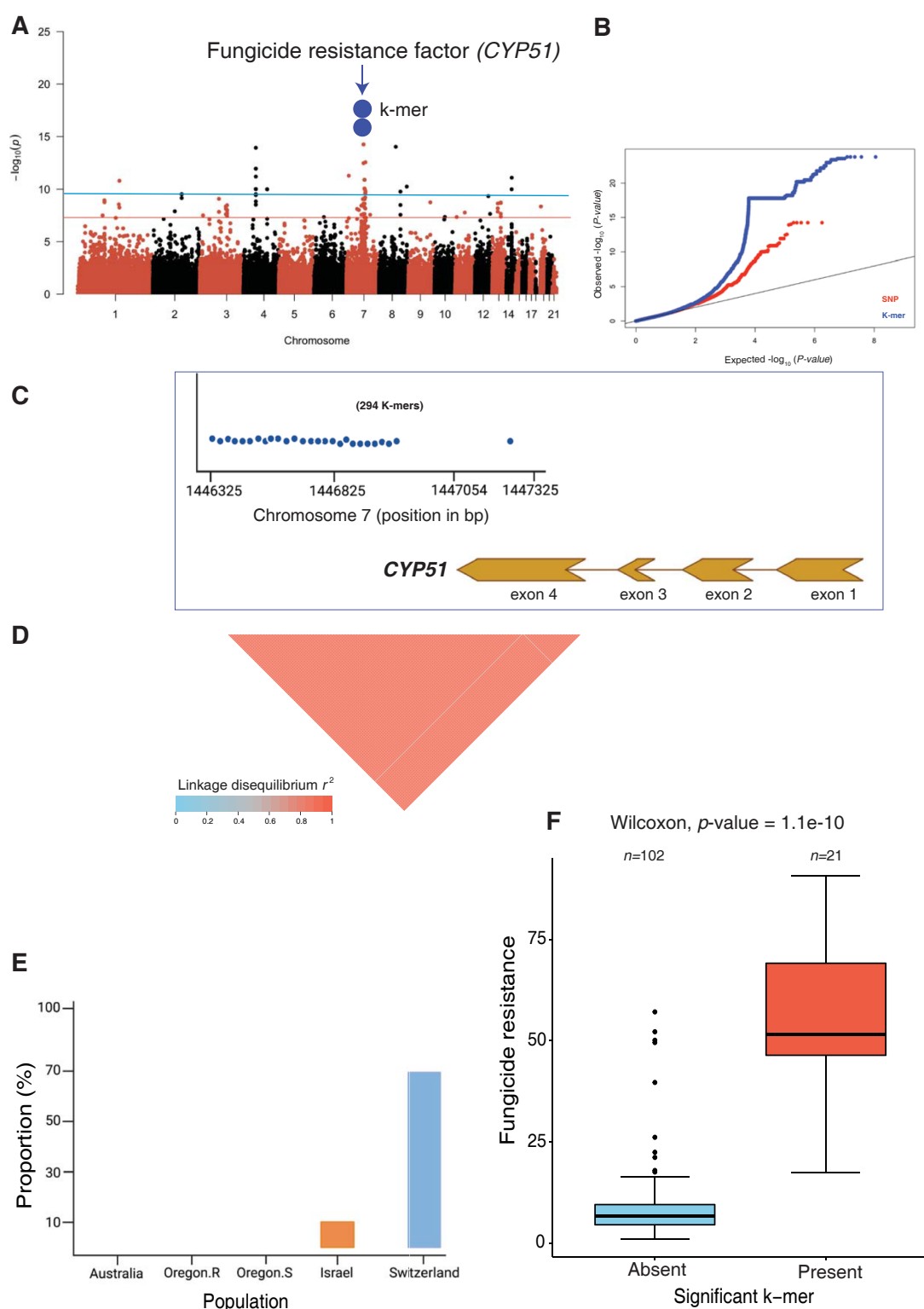

**Fig 5. Analysis of k-mer GWAS identifying causal genes underlying major phenotypes in *Zymoseptoria tritici*.** (A) Manhattan plot showing significant k-mers associated with fungicide resistance. The two blue dots represent all 294 significant k-mers with a unique genomic position on chromosome seven tagging the *CYP51* gene encoding the target of azole fungicides. The red and blue lines show the Bonferroni and permutation-based significance threshold ($\alpha = 0.05$) for SNP and k-mer GWAS, respectively. (B) Quantile-Quantile plot showing the *p*-value comparison between SNPs and k-mer based GWAS. (C) Physical

position of 294 significant k-mers mapped to unique positions on chromosome seven associated with the fungicide resistance gene *CYP51*. (D) Linkage disequilibrium (LD) heatmap showing the pairwise $r^2$ value among 294 significant k-mer presence/absence genotypes associated with the *CYP51* gene. (E) Proportion of isolates from different populations carrying significant k-mers that tagged *CYP51*. (F) Boxplot showing fungicide resistance levels in isolates with presence of the k-mers associated with the *CYP51* gene.

region of the gene spanning between the positions 1,446,325 and 1,446,893 bp. The k-mer presence/absence among isolates were in full linkage disequilibrium (**Fig 5C and 5D**; $r^2 = 1$). One additional k-mer localized (1,447,308 bp) to the fourth and largest exon of the gene and showed lower linkage disequilibrium ($r^2 = 0.48$) with the other k-mers. Most of the isolates from Switzerland (71.1%) and a few from Israel (10%) carried the k-mers associated with increased azole resistance (**Fig 5E** and **5F**). We expanded our analyses of k-mer associations to virulence traits (**Fig 6A**). We discovered 11 significant k-mers on chromosome 7 (from 1,897,941–1,897,951 bp) for virulence on the cultivar Runal. The tagged gene was previously identified through QTL mapping and encodes a virulence factor termed *Avr3D1*. No SNPs in the same region passed the Bonferroni significance threshold (**Fig 6A**). All k-mers were located in the largest exon and all but one was in full linkage disequilibrium with each other (**Fig 6C and 6D**). The k-mer with lower linkage disequilibrium to the other k-mers was primarily detected in isolates of the Israel population (**Fig 6E**). The isolates carrying the significant k-mers produced less leaf damage (**Fig 6F**).

## Discussion

Here, we report the most comprehensive assessment of association mapping performance to date for a plant fungal pathogen to unravel genetic determinants of phenotypic trait variation. We find that expanding association mapping to include multiple reference genome SNP data-sets provides a near linear increase in the number of additional loci detected by GWAS. Performing a reference-free GWAS approach using k-mers similarly boosted the power to uncover genetic variation underlying important traits. The extensive gains in the power of GWAS analyses that take into account structural variation reveals a greater proportion of the complexity inherent in adaptive genetic variation within microbial species.

SNP-based GWAS based on a single reference genome dataset have been successful in describing the genetic basis of complex pathogen traits [14,17,56,57]. By expanding the number of reference genome SNP datasets used for GWAS, we identified substantially more independent loci than what was previously identified using the same phenotype dataset [51]. The number of loci associated with most trait variation increased almost linearly with the addition of reference genome SNP datasets. Such an increase is striking given the fact that most traits are thought to be significantly constrained by stabilizing selection and have a conserved genetic basis (*e.g.* growth, melanization, reproduction) [58–60]. Stabilizing selection tends to reduce shared additive genetic variation between populations and closely related species, which ultimately reduces phenotypic variation [61]. Pathogen trait expression is expected to stabilize at an optimal level due to genetic trade-offs [51,62]. Climatic conditions and host genotype turnover may lead to rapid shifts in selection pressures. Hence, there should also be turnover in the genes underlying adaptation to previous environmental conditions. The *Z. tritici* pathogen model may be an outlier given the maintenance of very large population sizes, high gene flow and extensive chromosomal polymorphism [19,50]. The sub-linear increase in associated loci may also be explained, at least in part, by the use of a highly diverse, global panel of reference genomes. The reference genome isolates originating from six continents stem from populations that likely experienced divergent selection pressure from locally

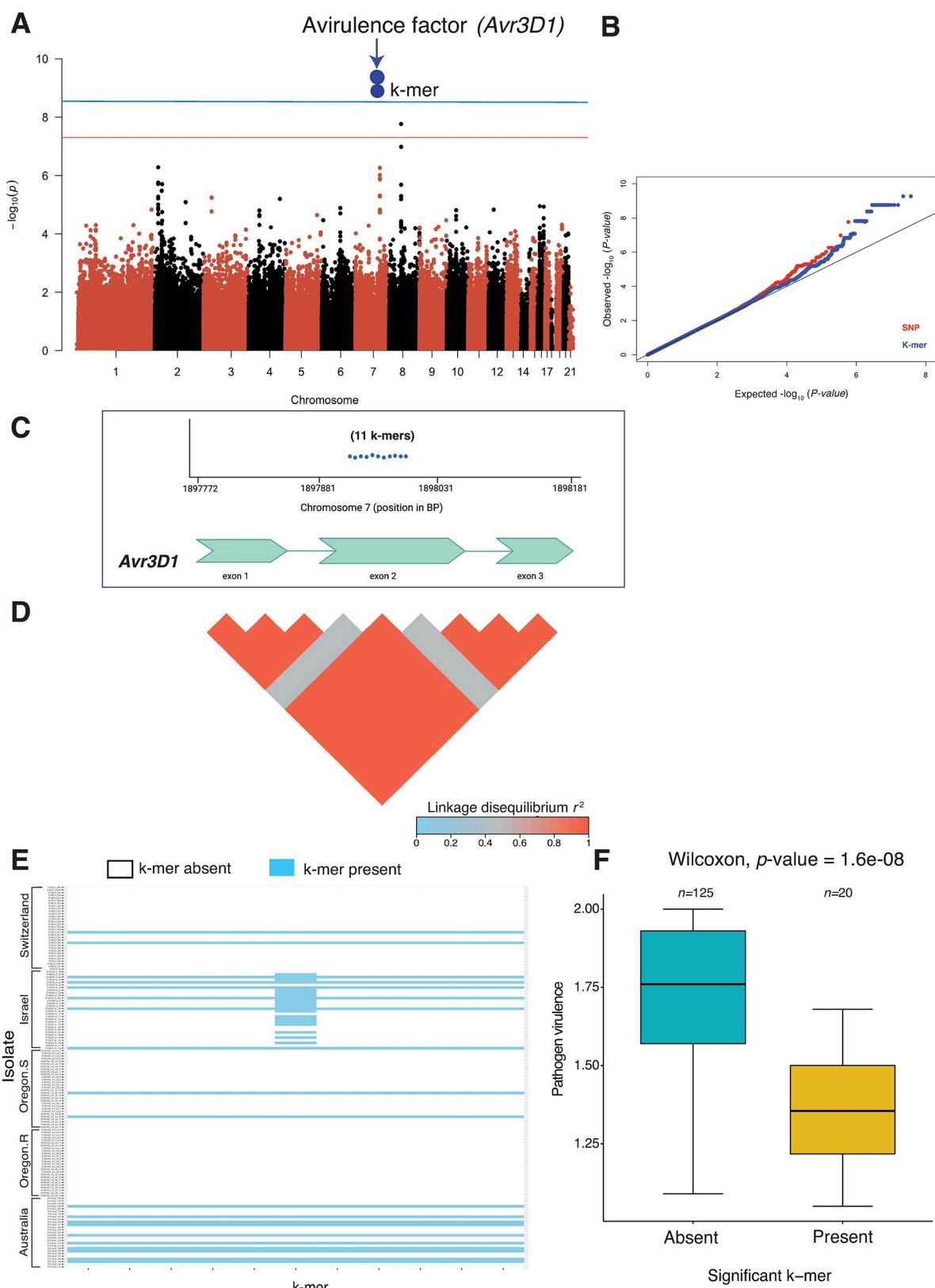

**Fig 6. k-mer based GWAS recovered a known effector gene in *Zymoseptoria tritici* with a higher statistical power than SNP-based GWAS.** (A) Manhattan plot showing significant k-mers associated with pathogen virulence on the wheat cultivar Runal. The two blue dots

represent all 11 k-mers uniquely mapping to positions on chromosome seven and tagging the avirulence gene *Avr3D1* encoding an effector protein. The red and blue lines indicate the Bonferroni and permutation-based significance threshold ($\alpha = 0.05$) for SNP and k-mer GWAS, respectively. (B) Quantile-Quantile plot showing the *p*-value comparison between SNPs and k-mers. (C) Physical position of 11 uniquely mapped k-mers on chromosome seven associated with *Avr3D1*. (D) Linkage disequilibrium (LD) heatmap showing the pairwise $r^2$ value among 11 significant k-mers associated with *Avr3D1*. (E) Presence/absence pattern of 11 significant k-mers associated with *Avr3D1* in five *Z. tritici* populations. The continuous horizontal blue line indicates isolates containing all the significant k-mers. (F) Boxplot showing pathogen virulence (percentage of the leaf surface covered by necrotic lesions) on the wheat cultivar Runal in isolates with or without the significant k-mers associated with *Avr3D1*.

adapted hosts and local climatic conditions. However, we found no meaningful variation in SNP-based heritability estimates depending on the reference genome being used, suggesting that the choice of the reference genome has little impact in this organism. Overall, we show that including a broad set of reference genome SNP datasets efficiently overcomes limitations imposed by using a single reference genome. Such limitations often stem from ascertainment bias in SNP calling and genetic distance between the reference genome and mapping populations [63]. A particular concern is that a single reference may not represent the full catalog of gene functions relevant for adaptation in the species pool [32]. For instance, missed associations for genes that are absent from a reference genome may underpin an adaptive advantage in a specific ecological context and/or geographic region [64,65].

It is important to note that the multiple-reference GWAS integrating multiple SNP panels introduces a significant statistical challenge. SNPs discovered based on a particular reference genome likely overlap with SNPs discovered based on other reference genomes. Performing GWAS on each SNP panel independently generates multiple testing issues as many SNP are likely tested for the same trait multiple times. This inherent dependence in the dataset increases the risk of false positives as significance thresholds are inadvertently set too leniently. These issues are particularly relevant when numerous SNPs are in strong LD and when a trait is governed by several small effect loci, as observed in this study. Accurately correcting for multiple testing would require a systematic investigation of homologous SNPs discovered across reference genomes. Such basepair-resolution in mapping polymorphism across multiple reference genomes is however challenging if not impossible for some divergent genomic regions. Pangenome graphs may provide a unified representation of genetic variation across multiple genomes, allowing for a consolidated view of SNPs without the need for multiple independent SNP sets. By mapping SNPs onto a pangenome graph instead of individual reference genomes, one could capture the near-complete genetic diversity represented by the reference genomes. Such pangenome mapped SNPs could then be subjected to a standard Bonferroni or FDR correction. Hence, the pangenome graph could streamline the GWAS and accurately provide multiple testing correction. Yet, pangenome graph construction algorithms remain a field of active investigation.

We find that accounting for genetic variation using k-mers instead of SNPs explains more genetic variation (*i.e.* gives a higher heritability). This implies that significant phenotypic variation is explained by genetic factors located in genomic regions that are difficult to access using SNPs. Such genetic variants are likely to be found in non-coding and TE-rich regions as indicated by the numerous significant k-mers tagging TE loci. It is important to note that k-mers may distinguish marginally different copies of the same TEs in the genome as distinct variants. Given the recent expansion in the number of TEs in the species generating many near identical TE copies of the same TE family [66], k-mers are likely still insufficient to localize associations to individual TE copies in the genome. Long-read sequencing of at least a subset of the GWAS panel is likely the only feasible solution to accurately define TE-generated genetic variation among strains. Despite the challenge in accurately capturing TE-associated variants, k-mers

tables established with quality control steps as implemented in this study should provide a reasonable balance between capturing genomic complexity and minimizing false positives. Such variants may be in accessory genomic regions absent from the reference genome and not easily accessed through SNP calling.

Missing heritability in human traits has been recovered by including rare genetic variants [67]. We show that incorporating genetic variants other than SNPs in plant pathogen GWAS increases trait heritability as well. We also found k-mers in extremely polymorphic regions of the core genome such as the regions surrounding the genes *CYP51* and *Avr3D1*. Recent studies have shown that SVs such as chromosomal rearrangements and copy number variations contribute to adaptive evolution in pathogens [68–70]. The k-mer approach broadly revealed three classes of loci: (1) loci previously identified by SNP-based GWAS, (2) gene functions that were not identified through SNP-based GWAS but have independent evidence for their contribution to phenotypic trait variation (*i.e. CYP51* and *Avr3D1*) and (3) previously unknown gene functions including genes encoding effector candidates for host manipulation and genes encoding detoxification functions (*e.g.* MFS transporters). The k-mer approach for GWAS has been successfully implemented for plants [35] and bacteria [18,71]. Here we provide strong evidence that such reference-free GWAS can also be successfully performed in eukaryotic microbial pathogens. The inclusion of all variant classes by the k-mer approach and utilization of k-mer presence/absence data can circumvent challenges associated with SNP variant calling to produce a less biased reflection of genetic variation. Although k-mer GWAS is relatively straightforward and computationally efficient, interpreting the k-mer associations is challenging due to the difficulties inherent in locating k-mer origins in the genome, assembling significant k-mers into contiguous local sequences, and ultimately associating k-mer variation with biologically meaningful sequence variation. While previous studies have mapped subsets of significantly associated k-mers to a reference genome [35,38], this approach may not accurately identify genomic locations of the k-mers, requiring additional analyses to capture the full sequence context. A largely unresolved concern with k-mer GWAS is a multiple testing issue similar to the one presented by multi-reference genome SNP panels. The redundancy arises because a single genetic variant can be tagged by multiple overlapping k-mers as seen in the case of *CYP51*, leading to correlated tests. As a result, the effective number of independent tests is often less than the total number of k-mers, making the Bonferroni threshold potentially too stringent. Hence, we used the permutation based threshold in k-mer GWAS as proposed previously [35].

Genetic variation in plant pathogens is characterized by high degrees of functionally relevant polymorphism as well as genomic plasticity underpinning accessory genes [19,72,73]. Beyond this, we found substantial complexity in the genes underlying the expression of the same trait under different environmental conditions. Working with such highly diverse pathogen populations poses serious challenges for selecting appropriate reference genome resources. Here we show that GWAS conducted on multiple reference genome SNP datasets and using reference-free approaches effectively compensates for this genetic diversity. This is supported by our recovery of known causal loci for specific phenotypes, including loci missed by previous GWAS, as well as a general improvement in heritability for all traits. Further refinements of our approach should integrate recent developments such as pangenome graphs that might alleviate limitations of studies based on SNPs and single reference genomes. Leveraging a multitude of GWAS signals following our combinatorial approach while controlling for multiple testing is likely to significantly advance our mechanistic understanding of pathogen emergence and adaptation.

## Material and methods

### Fungal material

A collection of 145 *Z. tritici* isolates sampled independently from four different wheat fields was used in this study. The field isolates were sampled between 1990 and 2001 from four different countries [74]: Australia (*n* = 27), Israel (*n* = 30), Switzerland (*n* = 32) and USA (Oregon.R, *n* = 26; Oregon.S, *n* = 30). The two Oregon populations were sampled from the wheat cultivar Madsen (moderately resistant) and Stephens (susceptible), growing simultaneously in the same field. Clones were removed from the field populations so that the analyzed panel comprises only strains with unique genotypes. Blastospores of each isolate were preserved in either 50% glycerol or anhydrous silica at −80˚C.

### Phenotyping for host infection traits

Datasets on virulence and reproduction for each pathogen strain were previously established by [75] (**S1F Table**). Virulence and reproduction were measured on 12 genetically different wheat cultivars displaying varying degrees of resistance and susceptibility to STB. The wheat panel included six commercial varieties (Drifter, Gene, Greina, Runal, Titlis, Toronit), a backcross line (Arina*Lr34*) and five landraces (1011, 1204, 4391, 5254, Chinese Spring). Four of the landraces (1011, 1204, 4391, 5254) came from the Swiss National Gene Bank (www.bdn.ch). Detailed phenotyping protocols are described in [75]. Briefly, three seeds of each cultivar were planted in a six-pot strip arrayed in a 2 × 3 pattern. Due to space limitations, the experiment was conducted in two stages, each including six cultivars. All plants were maintained in a greenhouse chamber at 22 ˚C (day) and 18 ˚C (night) with 70% relative humidity (RH) and a 16-h photoperiod. Blastospores of each isolate were inoculated using an airbrush spray gun until run-off on two-week-old seedlings to initiate the infection process. In both stages, the inoculations were repeated separately three times to generate three biological replications in separate greenhouse chambers. All inoculated second leaves were collected between 19–26 days post inoculation (dpi) and fixed on QR-coded A4 paper for scanning. The scanned images were analyzed using automated image analysis (AIA; [76]) to generate quantitative data on the amount of damaged leaf tissue (*i.e.* virulence) and the density of pathogen fruiting bodies called pycnidia produced within the damaged area (*i.e.* reproduction).

### Phenotyping for growth and stress-related traits

In vitro traits comprised fungal growth rate (mm per day), thermal sensitivity, mean colony area, fungicide resistance and melanization measured at different temperatures with or without fungicide (**S1G Table**) following previously described phenotyping protocols [57,77,78]. Briefly, after revival from long-term storage, each isolate was cultured on Petri dishes filled with yeast malt sucrose agar (4 g/L yeast extract, 4 g/L malt extract, 4 g/L sucrose, 50 mg/L kanamycin) for 4–5 days at 18 ˚C. Blastospore solutions were diluted using sterile water to a final concentration of 200 spores/ml using KOVA counting slides (Hycor Biomedical, Inc., Garden Grove, CA, USA). Petri dishes containing potato dextrose agar (PDA, 4 g/L potato starch, 20 g/L dextrose, 15 g/L agar) were inoculated with 500 μl of the spore solution. Inoculated plates were maintained at 15 ˚C (cold treatment) or 22 ˚C (control treatment) at 70% RH. Images were captured with a digital camera at 8, 11, and 14 days post inoculation (dpi) to generate five technical replicates. The photographs were analyzed using AIA macros in ImageJ as described in [77] to measure colony growth. The estimates of colony growth rate for each isolate were obtained by fitting a general linear model over three time points by taking the mean colony radii from 45 colonies. The growth rate ratio between colonies growing at 15 ˚C

or 22 ˚C, or on 22 ˚C PDA plates with or without propiconazole (Syngenta, Basel, Switzerland; 0.05 ppm) were expressed as temperature and fungicide sensitivity at 14 dpi, respectively. Fungicide resistance was also quantified on microtiter plates by growing 100 μl spore solutions at a concentration of $2.5 \times 10^4$ spores/ml of each isolate on 100 μl Sabouraud-dextrose liquid media (SDLM; 20 g/L dextrose, 5 g/L pancreatic digest of casein, 5 g/L peptic digest of animal tissue; Oxoid, Basingstoke, UK) with 12 different concentrations of propiconazole (0, 0.00006, 0.00017, 0.0051, 0.0086, 0.015, 0.025, 0.042, 0.072, 0.20, 0.55, 1.5 ppm propiconazole). Plates containing fungal spores amended with the fungicide of each isolate were gently shaken for one minute, sealed and incubated in the dark for four days at 22 ˚C with 80% RH. Three technical replicates of each isolate were performed. Fungal growth was estimated with an ELISA plate reader (MR5000, Dynatech) by examining the optical density (OD) at 605 nm wavelength. We estimated the $EC_{50}$ value (concentration at which the growth was reduced by 50%) for each isolate using dose-response curves across the varying fungicide concentrations using the drc v.3.0–1 package [79] in the R-studio (R Core Team, 2014). Furthermore, we estimated the ratio of colony area in fungicide environments (RCA_14_dpi_azole) to the control environment by utilizing the mean colony area measured for each isolate at 14 dpi on PDA at 15 ˚C (MCA_15 ˚C_14_dpi), 22 ˚C (MCA_22 ˚C_14_dpi), and 22 ˚C amended with 0.05 ppm propiconazole (MCA_14_dpi_azole) when compared to the control environment (RCA_14_dpi). Melanization of each isolate was measured at 8, 11, 14 and 18 dpi during growth at 15˚C, 22˚C and at 22˚C with 0.05 ppm propiconazole. We measured the mean gray value of fungal colonies from replicated plates for each isolate ranging from 0 (black) to 255 (white) for each time point. To provide a more intuitive interpretation of melanization, each mean gray value was subtracted from 255 to transform the original melanization scale to range from 0 (white) to 255 (black).

## Read mapping and single nucleotide polymorphism calling

We used publicly available raw Illumina whole genome sequences of 145 *Z. tritici* isolates (**S1H Table**; [51]). Trimmomatic v.0.36 [80] was used with the following settings (illuminaclip = TruSeq3-PE.fa:2:30:10, leading = 10, trailing = 10, slidingwindow = 5:10, minlen = 50) to trim off low-quality reads and remove adapter contamination from each isolate. Trimmed sequence data from all isolates were aligned to the *Z. tritici* reference genome IPO323 [46] using Bowtie2 v.2.3.3 with the option "—very-sensitive-local" [81]. We removed PCR duplicates from the alignment (.bam) files by using the MarkDuplicates module in Picard tools v.1.118 (http://broadinstitute.github.io/picard). Single nucleotide polymorphism (SNP) calling and variant filtration steps were performed using the Genome Analysis Toolkit (GATK) v.4.0.1.2 [82]. We performed SNP calling for all 145 *Z. tritici* isolates independently using the GATK HaplotypeCaller with the command "-emitRefConfidence GVCF; -sample_-ploidy 1" (*Z. tritici* is haploid). Then, GenotypeGVCFs was used to conduct joint variant calls on a merged gvcf variant file with the command -maxAltAlleles 2. SNPs found only in the joint variant call file were retained. As recommended by GATK Best Practices, we performed hard filtering of SNPs based on quality cut-offs using the GATK VariantFiltration and SelectVariants tools. Variants matching any of the following criteria were removed: QUAL < 250 (overall quality filter); QD < 20.0 (avoiding quality inflation in high-coverage regions); MQ < 30.0 (avoid calls from ambiguously mapped reads); $-2 > BaseQRankSum > 2$; $-2 > MQRankSum > 2$; $-2 > ReadPosRankSum > 2$; FS > 0.1. Using this procedure, the genotyping accuracy was shown to be high and congruent with an alternative SNP caller [50]. We retained a genotypic call rate of ≥80% and minor allele frequency (MAF) > 5% to generate a final SNP dataset containing 883,207 biallelic SNPs based on the reference genome IPO323.

We repeated the SNP calling and filtering procedure separately for 18 additional fully assembled *Z. tritici* genomes from [30]. The number of biallelic SNPs called on the 18 additional reference genomes ranged from 827,851 (genome TN09) to 883,119 (genome I93; **S1I Table**).

## SNP-based genome-wide association mapping

Log-transformed least-square means for each isolate × environment combination including 49 traits (**S1A Table**) were obtained from [75] to conduct genome-wide association (GWAS) mapping. Overall virulence was represented by calculating the average virulence of each isolate across 12 wheat cultivars. Reproduction specialization refers to the preference of isolates for specific hosts in order to optimize reproduction and was quantified using the adjusted coefficient of variation. This index was calculated as the logarithm of the adjusted variance divided by the mean of LSmeans for reproduction across all 12 hosts. We used a mixed linear model (MLM) approach implemented in the program GEMMA v.0.98 [83] to perform GWAS on all the traits. MLMs control for genetic relatedness and population structure [84,85]. Prior to GWAS, we converted all 19 SNP datasets (one per reference genome) into PLINK ".bed" format to perform principal component analyses (PCA) using the "—pca" command in PLINK v.1.90 [86]. To account for genetic relatedness among isolates, a centered genetic relatedness matrix (GRM) for each SNP dataset was constructed using the option "-gk 1" in GEMMA by considering all genome-wide SNPs. As both PCA and GRM can efficiently control for *p*-value inflation, we estimated genomic inflation factors (GIF, λ; [87] to make decisions on whether PCs should be included in the GWAS models as covariates or not. The GIF for each trait was estimated as λ = M/E, where M is the median of the observed chi-squared test statistics and E is the expected median of the chi-squared distribution [88]. The distribution of all SNP effects follows a one degree of freedom chi-square distribution under the null hypothesis with a median of ~0.455, which can be inflated by discrepancies in allele frequencies caused by population structure, genetic relatedness, and genotyping errors. We computed SNP effects only for the canonical reference genome IPO323 due to the difficulties in matching homologous SNP positions between reference genomes, hence direct comparisons between SNP effects estimated for different reference genomes would not be feasible. The inflation is proportional to the deviation from the null hypothesis. When the fitted GWAS model efficiently accounts for such systematic deviations, the λ value is close to 1. Therefore, depending on the λ value, the reference genome based GWAS were performed using either LMM+K or LMM+K+PC, where K is the GRM used as a random effect and the first three PCs were used as fixed covariates. We used the following LMM model in GEMMA:

$$y = W\alpha + x\beta + u + \varepsilon; u \sim MVN_n(0, \lambda\tau^{-1}K), \varepsilon \sim MVN_n(0, \tau^{-1}I_n)$$

where *y* represents a vector of phenotypic values for *n* individuals; *W* is a matrix of covariates (fixed effects with a column vector of 1 and the first three PCs), *α* is a vector of the corresponding coefficients including the intercept; *x* is a vector of the genotypes of the SNP marker, *β* is the effect size of the marker; *u* is a vector of random individual effects; *ε* is a vector of random error; $\tau^{-1}$ is the variance of the residual errors; *λ* is the ratio between the two variance components; *K* is the $n \times n$ genetic relatedness matrix and $I_n$ is an $n \times n$ identity matrix and $MVN_n$ represents the multivariate normal distribution. We set the MAF threshold to 5%. SNP *p*-values were estimated following a likelihood ratio test in GEMMA. We used the stringent Bonferroni threshold (*α* = 0.05; *p* = *α* / total number of SNPs) to define a SNP significantly associated with a phenotype. The proportion of phenotypic variance explained by the most significant SNPs was estimated by *2f(1-f)a²*, where *f* is the minor allele frequency and *a* is the standardized coefficient [89]. To obtain the standardized coefficient for each SNP, we estimated the

standardized regression coefficient applying a linear regression model with the "standard-beta" option implemented in PLINK v.1.9. We restricted this analysis only to the canonical reference genome IPO323. To identify genes close to significantly associated SNPs in one of the reference genomes [30], we used the BEDtools v.2.29.0 [90] *closest* command. We further investigated patterns of linkage disequilibrium (LD) in the genomic regions with the most significantly associated SNPs. All possible SNP pairs in 5 kb windows were analyzed using the "—hap-r2" command in vcftools. To visualize the $r^2$ values, heatmaps for each locus were generated using the R package LDheatmap v.0.99–7 [91]. We created a heatmap summarizing the number of significant SNPs passing the Bonferroni threshold for each trait and each genome using the R package *pheatmap*. To investigate patterns of associated SNPs near genes of interest, we identified orthologous gene sequences in other reference genomes or used BLASTn to obtain genomic coordinates if no homolog was not annotated in a particular reference genome. Using the genomic coordinates and the respective VCF files for each reference genome, we visualized the presence/absence pattern of SNPs using GenotypePlot (https://github.com/JimWhiting91/genotype_plot) in R.

### k-mer based genome-wide association mapping

We performed k-mer based GWAS on all 49 traits in the panel of 145 *Z. tritici* isolates following the methodology described in [35]. This approach uses raw sequencing reads of specific length and was designed for settings where a reference genome is lacking or to account for structural variation. As recommended, we generated the k-mers using raw sequencing reads that were previously filtered for adapter sequences, and low-quality reads. Specifically, this ensured that the k-mers generated were very unlikely to include sequencing artifacts. No specific bacterial contamination check was performed as DNA used for sequencing was derived from pure cultures of *Zymoseptoria tritici*. Furthermore, the alignment of reads to the complete *Z. tritici* reference genomes should prevent erroneous down-stream processing of contaminant reads. k-mers of 25 bp length were counted with and without canonization, sorted and listed in a textual format for each isolate separately. k-mer canonization refers to storing k-mers and their reverse-complement for generating presence/absence patterns since these sequences are indistinguishable [35]. k-mer length has an impact on the number and accuracy of k-mers. For small genomes of the size of *Z. tritici*, 25-bp k-mers are recommended [35]. k-mers were filtered based on the presence/absence patterns among isolates with a 5% MAF and compressed into a presence/absence table for running GWAS. There were 55,758,186 unique k-mers generated from 145 isolates. Prior to GWAS, a GRM was estimated with EMMA (Efficient Mixed-Model Association) that comprised an identity-by-state (IBS) matrix under the assumption that each k-mer has a small, random effect on the phenotype. GWAS was performed by using an LMM+K model in GEMMA with the likelihood ratio test to estimate *p*-values. A k-mer was considered to be significant when the *p*-value passed the permutation-based threshold as described in [35]. The pairwise LD among significant k-mers for each trait was estimated by converting the k-mer presence/absence table containing all the k-mers into PLINK format and using the command "—r2" in PLINK. We attempted to map all the significant k-mers for each trait to the *Z. tritici* reference genome IPO323 using the short-read aligner bowtie v1.2.2 [92] with the command "-a—best–strata". As a sanity check, we also aligned all the significant k-mers for each trait to one of the Swiss reference genomes 1A5. We used the center position of the k-mer alignment to the reference genome as a coordinate to inspect nearby features using BEDtools. If no significant k-mer could be mapped to the reference genome, we retrieved the isolates carrying the specific k-mer and used the paired-end raw sequencing reads to detect the origin of the k-mer. These paired-end reads were then aligned to the canonical reference genome IPO323 using Bowtie2 v.2.3.3 [81].

## Heritability estimation using SNPs and k-mers

We estimated SNP-based heritability on multiple reference genomes and k-mer-based heritability following the same procedure described in [51]. Briefly, the phenotypic data of each trait and the GRM representing the additive effect of all genome-wide SNPs from the canonical reference genome IPO323 and k-mers were included in a genome-based restricted maximum likelihood (GREML) approach using the genome-wide complex trait analysis (GCTA) tool v.1.93.0 [93] to estimate heritability. GRMs for reference genome SNP datasets and the k-mer presence/absence table (converted into PLINK format) were estimated following a normalized identity-by-state method and fitted as a random factor in the model to estimate the proportion of phenotypic variance for each trait. The following formula from [93] was used to estimate the relatedness between two individuals:

$$A_{jk} = \frac{1}{N}\sum_{i=1}^{N} \frac{(x_{ij} - 2p_i)(x_{ik} - 2p_i)}{2p_i(1 - p_i)}$$

Where $x_{ij}$ is the number of copies of the reference allele for the $i^{th}$ SNP of the $j^{th}$ individual and $p_i$ is the frequency of the reference allele and $N$ is the number of SNPs. Here, the GRMs were constructed using all genome-wide SNPs and k-mers irrespective of the nature of their relationship with the phenotype, thus indicating the approximated genetic similarities at causal loci and the accuracy of the heritability estimates.

## Pangenome analyses

We generated accumulation curves to estimate the gain in additional loci from performing GWAS on more than one reference genome. We pursued analyses only for the set of traits for which significant GWAS associations were found. For this, we retrieved for each GWAS based on SNPs mapped to a particular reference genome the set of genes within 1 kb distance with significantly associated SNPs. Then, we matched the set of associated genes among genomes using within-species gene orthology information [30] to determine whether genes belong to the same orthogroup. We used a sampling procedure (without replacement) among reference genomes to assess the total number of distinct orthogroups with a significantly associated gene. The accumulation curves for 1–19 genomes were produced using the "specaccum" function in the R package *vegan* [94]. We fitted an Arrhenius nonlinear model to the gene accumulation curve to visualize the distribution using the "random" and "fitspecaccum" commands. UpSetR package [95] was used to visualize the number of significantly associated genes identified by the multiple reference-based GWAS and k-mer GWAS. All other figures were generated using the R packages *qqman* [96] and *ggplot2* v.3.1.0 [97].

## Supporting information

**S1 Fig. Genomic inflation factor estimated from genome wide association mapping using principal components as covariates and without principal components. (A)** host-related traits *i.e.* pathogen virulence (percentage of the leaf surface covered by necrotic lesions) and reproduction (pycnidia density within lesions) and **(B)** environmental stress related traits. Pathogen virulence and reproduction were measured on 12 genetically diverse wheat lines. (PDF)

**S2 Fig. Quantile-quantile plot of *P*-values for GWAS with and without principal components as covariates in different traits for the reference genome IPO323.** Red dots indicate *P*-values estimated with the first three principal components as covariates in the GWAS and

blue dots indicate *P*-values estimated without principal components.
(TIFF)

**S3 Fig. SNP polymorphism pattern surrounding a target gene that is tagged with a significant SNP association.** For **(A)** reference genome KE94 and **(B)** reference genome IPO323. The color gradient indicates the estimated allele frequency of each SNP in the respective position. A frequency of 0 indicates the allele is absent and 1 indicates the allele is fixed.
(PDF)

**S4 Fig. Comparison of heritability estimates based on SNPs (for the reference genome IPO323) and K-mers. (A)** Pathogen reproduction (pycnidia density within lesions), **(B)** pathogen growth rate and fungicide resistance, **(C)** pathogen melanization. Pathogen reproduction was measured on 12 genetically diverse wheat lines. Overall reproduction represents the average value of reproduction measured on 12 genetically diverse wheat lines. Reproduction specificity was estimated based on the adjusted coefficient of variation of mean reproduction across 12 genetically diverse wheat lines. Higher specificity suggests affinity to certain hosts for maximizing reproductive fitness. Both SNP-based and K-mer-based heritability were estimated by following a genome-based restricted maximum likelihood (GREML) approach. Standard errors are indicated by error bars. **(D)** Alignment of significantly associated K-mers against the reference genome (IPO323) show the proportion of K-mers having a unique mapping position, multiple locations, or no unambiguous mapping position in environmental stress-related traits. **(E)** Proportion of significant K-mers with a unique mapping position in the reference genome either tagging a gene or a transposable element in environmental stress-related traits.
(PDF)

**S5 Fig. Alignment of significantly associated k-mers against the reference genome 1A5.** Proportion of k-mers having a unique mapping position, multiple locations, or no unambiguous mapping position in **(A)** Host related and **(B)** environmental stress-related traits.
(PDF)

**S6 Fig. Upset plots showing the number of genes identified in pathogen virulence (percentage of the leaf surface covered by necrotic lesions), reproduction (pycnidia density within lesions), fungicide resistance and pathogen melanization that are unique or shared among 19 reference genomes in *Zymoseptoria tritici* and the K-mer approach.** The intersection size is the number of genes and the black dots on the matrix represent whether the genes are shared or unique to different reference genomes and the K-mer approach. For example, the first vertical bar in each graph shows the number of genes that are uniquely identified by the K-mer GWAS, while the last vertical bar demonstrates the number of genes that are commonly identified by all the reference-based and K-mer GWAS. Pathogen virulence and reproduction were measured on 12 genetically diverse wheat lines.
(PDF)

**S1 Table. S1A Table Raw phenotypic data for virulence (measured as the amount of necrotic lesion area) and reproduction (pycnidia density within lesion area) on 12 wheat cultivars from 145 *Zymoseptoria tritici* isolates. S1B Table: Raw phenotypic data for mean colony area per plate and mean grey value per plate measured in different temperatures and in presence/absence of fungicide from 130 *Zymoseptoria tritici* isolates.** "NA" indicates that no data were obtained due to no colony growth or contamination. **S1C Table: Description of 145 *Zymoseptoria tritici* isolates with their corresponding sampling location, year and NCBI SRR Run ID for the whole genome sequence data used in this study. S1D Table:**

**Number of single nucleotide polymorphisms (SNPs) called on 19 reference genomes at 5%
minor allele frequency and 80% genotyping rate for 145 *Zymoseptoria tritici* isolates. S1E
Table: List of phenotypic traits used for GWAS in this study. S1F Table: Number of signifi-
cant SNP associations above the 5% Bonferroni significance threshold for 20 traits com-
prising pathogen virulence, reproduction and environmental stress mapped in 19
reference genome SNP datasets. S1G Table: Summary statistics of genome-wide SNPs
passing the Bonferroni significance threshold of 5% for specific traits identified 19 refer-
ence genome SNP datasets.** SNPs are ordered according to the smallest P-value. **S1H Table:
List of genes with their predicted protein functions in close proximity ($<$ 1 kb) to signifi-
cant SNPs above the Bonferroni significance threshold (alpha = 0.05) across 19 reference
genome SNP datasets for different traits of *Zymoseptoria tritici* isolates. S1I Table. List of
genes with their predicted protein functions in close proximity ($<$ 1 kb) to significant k-
mers above the permutation-based significance threshold (5%) for different traits of
Zymoseptoria tritici isolates.**
(XLSX)

## Acknowledgments

Emile Gluck-Thaler provided helpful comments on a previous version of the manuscript.

## Author Contributions

**Conceptualization:** Anik Dutta, Daniel Croll.

**Formal analysis:** Anik Dutta.

**Funding acquisition:** Bruce A. McDonald.

**Investigation:** Anik Dutta.

**Supervision:** Bruce A. McDonald, Daniel Croll.

**Writing – original draft:** Anik Dutta, Daniel Croll.

**Writing – review & editing:** Bruce A. McDonald, Daniel Croll.

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
