## [Decision Letter · Decision Letter 0]

24 Jul 2023

Dear Prof Croll,

Thank you very much for submitting your manuscript "Combined reference-free and multi-reference based GWAS uncover cryptic variation underlying rapid adaptation in a fungal plant pathogen" for consideration at PLOS Pathogens. As with all papers reviewed by the journal, your manuscript was reviewed by members of the editorial board and by several independent reviewers. In light of the reviews (below this email), we would like to invite the resubmission of a significantly-revised version that takes into account the reviewers' comments.

We cannot make any decision about publication until we have seen the revised manuscript and your response to the reviewers' comments. Your revised manuscript is also likely to be sent to reviewers for further evaluation.

Sincerely,

David E. Cook, PhD

Guest Editor

PLOS Pathogens

Bart Thomma

Section Editor

PLOS Pathogens

Kasturi Haldar

Editor-in-Chief

PLOS Pathogens

orcid.org/0000-0001-5065-158X

Michael Malim

Editor-in-Chief

PLOS Pathogens

orcid.org/0000-0002-7699-2064

Dear Dr. Croll,

The submitted manuscript has undergone peer review by three independent experts. The reviewer’s all commented on the value of the analysis, the demonstration of bias in single-reference GWAS studies, and that while the approach is not entirely new, it is a good demonstration and guide-post for use in fungal systems.

There are questions raised by the authors that require addressing and warrant the decision of major revision. Some reviewers suggested the revisions could be considered minor, which can be interpreted that the authors did not think major new analyses are necessary, but that more explanation, justification, and discussion are warranted. Please pay attention to the comment by reviewer 2 asking for specific examples demonstrating PAV as causative for trait association differences, and comments by all authors regarding statistical approaches and justification.

Reviewer's Responses to Questions

**Part I - Summary**

Reviewer #1: The manuscript entitled “Combined reference-free and multi-reference based GWAS uncover cryptic variation underlying rapid adaptation in a fungal plant pathogen” collected 49 phenotypic traits of 145 strains of the fungal wheat pathogen Zymoseptoria triticim, and genotyped these strains through whole genome sequencing. Genome-wide association studies (GWAS) were conducted using a traditional SNP-based approach along with multiple reference genomes and a k-mer method to analyze various collected traits. The results revealed significantly associated SNPs, k-mers, and candidate genes. While GWAS has been extensively utilized in genetic research involving humans, animals, and plants, its application in microorganisms is relatively limited. Therefore, the utilization of GWAS, particularly k-mer GWAS given the prevalence of structural variation among Z. triticim fungal genomes, holds significant value. This study devoted significant efforts to data collection, and the methods employed for data analysis are well-described, providing ample detail. A few comments are provided for the authors' consideration.

Reviewer #2: Dutta et al. report the application of innovative association mapping techniques to discover hidden variation in Zymoseptoria tritici associated with virulence/ecological traits. As fungal pathogens often have highly variable genomes within a species including structural variation, classic GWAS methods that rely on SNPs called from a single reference genome inherently miss much of the standing variation, which can fail to detect true genetic associations. This manuscript takes a comprehensive approach using a vast phenotypic dataset to assess the use of SNPs called from multiple reference genomes, as well as a reference-free k-mer approach. Overall, the manuscript is very well written. The methods used are clearly described and the conclusions are generally well-founded. These results advance our understanding of functional genomic variation in fungi and provide a framework that can be applied in other systems. I have identified a couple of areas that could be expanded and some relatively minor points that I feel could bolster some conclusions and increase the clarity of the results.

Reviewer #3: The authors use a variety of their previously published population-level data from Z. tritici (isolate WGS, several reference genome assemblies, phenotype data) in order to address an important issue - for species with abundant structural variation and differences in gene and marker presence-absence, how much is missed in association mapping studies when defining markers solely based on a single reference genome? They compare the performance of GWAS using SNPs called with respect to the single canonical reference genome to GWAS with SNPs called with respect to an additional 18 reference genomes as well as to GWAS using a reference-genome free k-mer approach. Not surprisingly, they find that the reliance on just a single canonical reference genome generally leads to missing the identification of other significant trait-associated markers that are absent or cannot be called with the canonical reference. The k-mer GWAS approach has already been demonstrated in plant and bacterial species, this is a novel and significant demonstration of the utility of this approach in fungi, in particular in an important plant pathogenic fungus.

The manuscript is generally well-written and executed, though some areas require some clarification or revision (see below). A strength of the study is the availability of so much data from their previous studies, such that this study can focus on a comparison of analysis methods and remain fairly concise. The authors do a great job of providing important data in supplemental tables so the reader can follow along with their conclusions well. The most significant weaknesses - described immediately below - are probably something the authors can address simply - either by providing a little more information and interpretation or a bit of justification of some choices they have made in their analyses.

The manuscript refers in the abstract and other places to a set of 49 traits for GWAS, but then results of GWAS are only shown for 20 or so different traits. It is not clear why this is so, and should be explained - the biggest concern without any additional explanation is that the authors chosen subset of traits are not representative of the full results in some important aspects that could affect the study's conclusions. (more related comments on this issue in PartII of review).

I agree with the authors on the relative merits of the k-mer approach in contrast to performing GWAS separately against multiple SNP sets each called against a different reference. The latter approach clearly entails a lot of redundancy (most of the SNPs are presumably redundant across most of the SNP sets). The other major weakness in the study has to do with the tough statistical question related to the approach of using multiple SNP sets. How to account for multiple testing when performing GWAS with the same trait data and not just 1 set of SNPs called against 1 reference, but 18 additional SNP sets called against different references? If the additional SNPs were independent (and they clearly are not), then defining 5% significance with Bonferroni correction would mean dividing the original p-values not only by the number of SNPs in each set (>800k) but also by the total number of SNP sets (19). In reality, there should be a very high correlation in SNP sets (as fig S3 generally shows), but still the additional tests that are actually being performed with each new SNP set are not properly being taken into account by the Bonferroni correction. Simply, with each SNP set we expect some small amount of false positives, and by doing 19 sets of tests, we of course expect some increase in false positives, even if small.

**Part II – Major Issues: Key Experiments Required for Acceptance**

Reviewer #1: (No Response)

Reviewer #2: Although I agree with the authors that the differential detection of loci depending on the reference genome used is likely due to presence/absence variation or issues with reliably calling SNPs, I feel that the results could be bolstered by investigating this further. Are the genes in close proximity to these reference-genome specific loci truly absent in the isolates where the locus was not detected? Or if the gene is present in reference datasets where the locus was not detected, were those regions devoid of SNPs and/or aligned reads? Choosing a few of these scenarios and exploring them further may help illustrate the impact of these results a little more clearly.

L221-234: Why limit the k-mer mapping to the IPO323 genome? One of the major novelties of this manuscript is capturing and associating unexplored genetic variation to phenotypic traits and by only mapping significant k-mers to IPO323, I feel that some important associations to novel genes may be missed. Would mapping the identified k-mers to the alternate references help place some of the unmapped k-mers? Or could k-mers be assembled into contigs and used to search the pangenome gene space? I think this would help further demonstrate the utility of this approach if all available data can be used and would also more accurately reflect the proportions of mapped/unmapped k-mers.

Reviewer #3: As mentioned above, the authors state using 49 traits, but although 2 supplemental tables are used to provide the raw measurements taken for the traits, the full list of 49 traits are never properly listed anywhere (even in a supplemental table). In the methods they describe how some of the trait values are calculated based on the raw measurements taken, but without the full list of 49 traits I do not know if the description of calculating any other traits is missing. Please include a full listing of the 49 traits.

Related - what is EAM (line 239 and in some supplemental tables) - is this the same as MCA? And overall virulence or overall reproduction, and reproduction specificity - these are derived phenotypes mentioned later in the manuscript but not otherwise defined within the Methods section of the paper.

And related to the number of traits, the authors focus their conclusions (including figures and supplemental tables) on a subset of only 20 traits, but it is never explained why - some additional explanation here is required!

I found it odd that in table S5 there are no cases where 0 significant SNPs were recovered across all of the different reference genomes. Does this mean the authors filtered out their results to drop any cases where 0 significant SNPs were recovered in 1 or more of the reference genome analyses? Or traits were dropped because no significant SNPs were found in ANY of the reference genome analyses? Were other traits dropped, and if so, for what reason? And could any of the study's conclusions change if additional traits were included in the paper?

Please address this issue.

The other major issue requires some discussion of multiple testing and the Bonferroni corrections used, in the context of their study design. For the most part, the value of the study is in the comparisons of the results when performing 1 individual GWAS analysis (say, just k-mer, or SNP based using only 1 particular reference) versus the results when performing a different GWAS analysis. In this case, the Bonferroni corrections are probably fine. But by suggesting that other researchers might take their approach of defining multiple SNP sets and performing GWAS across all of them in order to maximize the number of significant markers identified is statistically more problematic - here is where the Bonferroni correction as carried out is probably not quite adequate, or at least misleading, and this needs to be stated somewhere. Perhaps there is some other way (independent of the k-mer approach) to combine SNP markers across the different SNP sets (pangenome graphs, as mentioned in line 336?) that would avoid so much redundancy in tests performed, where the total number of tests could be reduced and a better/more conservative Bonferroni correction could be applied to help reduce the chance of false positives. I am not asking the authors to alter their analyses methods in this paper to address the issue, but at a minimum they must point out the issue, so that other researchers would know not only is using multiple reference assemblies as they have done costly in effort (making the assemblies and defining each SNP set), but it would also be more statistically costly too, because it does involve performing (many) more tests.

**Part III – Minor Issues: Editorial and Data Presentation Modifications**

Reviewer #1: 1. First, I noticed that two different statistical approaches were used to account for multiple tests in GWAS. In SNP GWAS, a very stringent approach, the Bonferroni method, was used, while a permutation approach was employed for the k-mer GWAS. The Bonferroni correction method assumes that each marker is independent, which is not true and may limit the power of SNP GWAS. I am curious why the permutation approach cannot be used as well for SNP GWAS.

2. Three reasons were provided to justify the selection of 25-mer (L197-200). I think the selection of the k-mer size is related to the size of the complexity of genomes. 25-mer seems to be a reasonable selection for the small microbial genome. However, even for small genomes, the longer k-mer can facilitate accurate genomic mapping to find k-mers’ genomic location. In the justification, Reason 2 is a little hard to understand. Sequence sharing should not be a big issue among closely related genomes since the density of genomic polymorphisms among them should be low. In addition, given the high accuracy of Illumina read, sequencing errors mentioned in Reason 3 are not a big issue either as long as the k-mer length is not too long. Maybe the justification can be revised.

3. Trait-associated k-mers are frequently found in TE. Because presence/absence k-mers were used as markers, here are possibilities: 1) An associated TE is present in some strains but in others; 2) A TE have many copies in all genomes (or strains) and a particular copy carrying variants was associated with traits. For the latter case, the TE copy carrying variants might insert to certain genes for phenotypic impacts. This is likely to be hard to investigate but it would be nice if more details can be provided.

Minor comments:

4. Although genomic inflation factors were calculated and controlled, if I understand this correctly, the p-value inflation based on Q-Q plots in Figure 4 seems to be high. Please help explain.

5. Of p-values in Figure 5F and Figure 6F, it is probably not correct to make numbers after “e” be superscript, 1.6e-08 is very different from 1.6e(-08 as superscript).

Reviewer #2: L132-133: It is not clear if GIFs were calculated for each individual reference-based GWAS (each trait/SNP dataset combination).

L128-149: The variation observed among the various alternative reference genome datasets is very interesting. However, instead of reporting the number of SNPs, it would be informative to report the number of distinct loci. Presumably, multiple SNPs in LD are being detected at a single locus. I suggest delimiting these results further into the number of individual loci (by examining local LD or setting an LD threshold to bin SNPs) which may more accurately reflect the number of genes detected for each trait. Although the ortholog accumulation analysis somewhat gets to this point, I still think this information should be relayed when discussing the genetic analyses.

L483-484: Filtering for adapter sequences and quality were previously described, but how were the raw sequences filtered for contamination?

Figure 2B: The y-axes have different maximum limits. I suggest making the y-axis ranges uniform to make the visualization more comparable. Also, is the left Manhattan plot showing associations with virulence on landrace 1204? The specific line should be mentioned in the figure title and/or caption.

Figure S3 was blurry and difficult to read. Also, “Pathogen” is misspelled in the upper left quadrant.

-Some references in the Materials and Methods are cited in a different format (for example: Hartmann et al. 2018 in L426)

Reviewer #3: Please indicate in which analyses PCs were used as a factor or not! The Methods vaguely describes on what basis this decision was made, but it may be important in interpreting the results to know exactly when PCs were used or not.

Based on your plots, it appears the GIFs are averages across the different reference genomes? Please clarify this. Also, why not show some q-q plot comparisons, at least for a few examples of analyses using PCs and not using them - wouldn't this be more informative than only using GIF?

In table S5 and in the results - the variance of number of significant SNPs identified is reported (and emphasized). But of course we expect a higher variance when the mean is also higher. I suggest also computing the coefficient of variation (cv) in these cases, and reporting that instead. You will see that different traits end up being extreme when looking at cv instead of variance.

SNP effect size was only estimated for the canonical reference - why? I suppose it appears in the manuscript just for illustrative purposes, and these effect sizes are not expected to be very different against other references, but it seems maybe not difficult to estimate all effect sizes. If you keep effect sizes only for the canonical reference, please provide a justification.

Fig 3 please provide a statement that you are summing the distinct genes (or orthologous groups) across these sets of traits for the plot (not taking an average).

Fig 2B - list cultivar 1204.

Fig 4 - am I correct in assuming that the blue dots (significant k-mers) always come in pairs because they represent the alternative SNP alleles (now coded as presence-absence for discrete k-mers)? If so, it would help to explicitly explain this.

Lines 159-161: State that you are only counting a gene once even when it is significant for multiple (related) traits.

Fig S3 is cited before S2 - is that a problem for the journal?

In figure S2A and S3, the figure label reads pathoge rather than pathogen.

In Fig5B and line 241 - given the gene orientation, shouldn't the text read downstream rather than upstream? (with respect to gene's transcription?)

Line 251 - maybe cite Fig 6A not 6B? Because the q-q plot does not indicate from which region significant SNPs come.

If initial SNP filtering requires genotype call rate >80%, then why is the 50% maximum missing values filter needed later?

PLOS authors have the option to publish the peer review history of their article (what does this mean?). If published, this will include your full peer review and any attached files.

Reviewer #1: No

Reviewer #2: No

Reviewer #3: No
---

## [Editor Report · Decision Letter 1]

6 Nov 2023

Dear Prof Croll,

We are pleased to inform you that your manuscript 'Combined reference-free and multi-reference based GWAS uncover cryptic variation underlying rapid adaptation in a fungal plant pathogen' has been provisionally accepted for publication in PLOS Pathogens.

Best regards,

David E. Cook, PhD

Guest Editor

PLOS Pathogens

Bart Thomma

Section Editor

PLOS Pathogens

Kasturi Haldar

Editor-in-Chief

PLOS Pathogens

orcid.org/0000-0001-5065-158X

Michael Malim

Editor-in-Chief

PLOS Pathogens

orcid.org/0000-0002-7699-2064

Dear Dr. Croll,

The revised manuscript by you and your co-authors has adequately addressed reviewer comments. Congratulations on the improved manuscript, and I am happy to inform you that the manuscript has been recommended for acceptance.

---

## [Editor Report · Acceptance letter]

10 Nov 2023

Dear Prof Croll,

We are delighted to inform you that your manuscript, "Combined reference-free and multi-reference based GWAS uncover cryptic variation underlying rapid adaptation in a fungal plant pathogen," has been formally accepted for publication in PLOS Pathogens.

Best regards,

Kasturi Haldar

Editor-in-Chief

PLOS Pathogens

orcid.org/0000-0001-5065-158X

Michael Malim

Editor-in-Chief

PLOS Pathogens

orcid.org/0000-0002-7699-2064